# Machine Learning Approaches for Activity Recognition and/or Activity Prediction in Locomotion Assistive Devices—A Systematic Review

**DOI:** 10.3390/s20216345

**Published:** 2020-11-06

**Authors:** Floriant Labarrière, Elizabeth Thomas, Laurine Calistri, Virgil Optasanu, Mathieu Gueugnon, Paul Ornetti, Davy Laroche

**Affiliations:** 1INSERM, UMR1093-CAPS, Université de Bourgogne Franche Comté, UFR des Sciences du Sport, F-21000 Dijon, France; floriant_labarriere@etu.u-bourgogne.fr (F.L.); elizabeth.thomas@u-bourgogne.fr (E.T.); paul.ornetti@chu-dijon.fr (P.O.); 2PROTEOR, 6 rue de la Redoute, CS 37833, CEDEX 21078 Dijon, France; Laurine.Calistri@proteor.com; 3ICB, UMR 6303 CNRS, Université de Bourgogne Franche Comté 9 Av. Alain Savary, CEDEX 21078 Dijon, France; virgil.optasanu@u-bourgogne.fr; 4INSERM, CIC 1432, Module Plurithematique, Plateforme d’Investigation Technologique, CHU Dijon-Bourgogne, Centre d’Investigation Clinique, Module Plurithématique, Plateforme d’Investigation Technologique, 21079 Dijon, France; mathieu.gueugnon@chu-dijon.fr; 5Department of Rheumatology, Dijon University Hospital, 21079 Dijon, France

**Keywords:** machine learning, locomotion, assistive devices, embedded sensors

## Abstract

Locomotion assistive devices equipped with a microprocessor can potentially automatically adapt their behavior when the user is transitioning from one locomotion mode to another. Many developments in the field have come from machine learning driven controllers on locomotion assistive devices that recognize/predict the current locomotion mode or the upcoming one. This review synthesizes the machine learning algorithms designed to recognize or to predict a locomotion mode in order to automatically adapt the behavior of a locomotion assistive device. A systematic review was conducted on the Web of Science and MEDLINE databases (as well as in the retrieved papers) to identify articles published between 1 January 2000 to 31 July 2020. This systematic review is reported in accordance with the Preferred Reporting Items for Systematic reviews and Meta-Analyses (PRISMA) guidelines and is registered on Prospero (CRD42020149352). Study characteristics, sensors and algorithms used, accuracy and robustness were also summarized. In total, 1343 records were identified and 58 studies were included in this review. The experimental condition which was most often investigated was level ground walking along with stair and ramp ascent/descent activities. The machine learning algorithms implemented in the included studies reached global mean accuracies of around 90%. However, the robustness of those algorithms seems to be more broadly evaluated, notably, in everyday life. We also propose some guidelines for homogenizing future reports.

## 1. Introduction

Healthy humans are easily able to adjust locomotor pattern to deal with multiple environments encountered in daily living situations such as stair ascent/descent, slope ascent/descent, obstacle clearance, walking on uneven floors, cross-slopes or different surfaces. Hence, with lower limb impairments such as unilateral lower limb amputation, it becomes challenging to deal with most of these environmental changes [1].

To handle this issue, intelligent devices such as the C-leg TM (OTTOBOCK, Berlin, Germany) or the Rheo knee (ÖSSUR, Reykjavík, Iceland) have been developed. These variable-damping prostheses, compared to mechanically passive prostheses, improved the smoothness of gait, and decreased hip work during level-ground walking [2]. Additional improvement was provided by a powered prosthesis which was reported to decrease the metabolic cost of transport when compared to a conventional passive prosthesis in similar conditions [3]. Prosthetic devices which passively or actively mimicked human actions were found to be of help. One historic example of such innovation was the energy return foot that reproduced foot behavior and improved the gait of patients with amputation. Other innovations in the attempt to create intelligent devices can be seen with some microprocessor-controlled prostheses with the ability to recognize the terrain being traversed (e.g., Genium OTTOBOCK, Berlin, Germany, Linx BLATCHFORD, Basingstoke, UK). It only stands to reason that the next step in this progression would be the development of devices with the ability to make predictions for automatic gait adjustments across multiple terrains.

Developments in these efforts have come in the form of intelligent controllers on locomotion assistive devices. In such devices, gait is regulated by a hierarchical three-level controller [4]. The highest-level controller is responsible for detecting the user-intent. The mid-level controller automatically switches the control law (e.g., the powered active transfemoral prosthesis developed by Vanderbilt University [5]) of the device in accordance with the high-level controller output. The low-level controller compares the desired state of the device to the sensed state and corrects it when needed. The detection of user-intent is done either by the user directly communicating his intentions to the device using a controller, or by automatic interpretation by an algorithm. Examples of the first are the control buttons found in the ReWalk TM exoskeleton (ARGO MEDICAL TECHNOLOGIES Ltd., Yokneam, Israel) or predefined body movements which allow the wearer of the Power KneeTM (ÖSSUR, Reykjavík, Iceland) to switch between locomotion modes. In this device, switching between locomotion modes requires the user to stop or to perform certain unnatural body movements. As opposed to these explicit methods, algorithm-based implicit methods interpret user intent. Such algorithm-based techniques allow smoother transitions by automatically switching between the control laws of the device. A more promising approach might be one based on machine learning algorithms. Such algorithms automatically detect user-intent by mapping sensor data to an associated locomotion task.

There are numerous studies in which machine learning has been used to adapt the behavior of orthotic/prosthetic devices to user locomotion mode. We performed a systematic review that identifies and summarizes such studies. Under the scope of this review, reports were selected if (1) body-worn sensors or sensors embedded in the devices were used (2) machine learning classifiers were able to identify the investigated locomotion modes of human volunteers. It covers essential technical details such as the pre-processing methods which were used, the specific Machine Learning algorithms which were employed, and the corresponding accuracies obtained. By the end of this review we aim to propose recommendations for future studies and some suggestions concerning the uniformization of the terms used to report results in the field.

## 2. Material and Methods

This systematic review, registered on PROSPERO (CRD42020149352), is reported in accordance with the Preferred Reporting Items for Systematic reviews and Meta-Analyses (PRISMA) guidelines [6].

### 2.1. Eligibility Criteria

This systematic review included peer-reviewed articles and patents focusing on the Machine Learning (ML) approach for classifying locomotion modes in volunteers wearing assistive devices (see below for definition of the included devices). For this purpose:The algorithms must be based on locomotion data collected from embedded sensors in the device or from body-worn sensors. Studies evaluating a previously developed ML-based pattern recognition algorithms were also included. We focused on Machine Learning methods that carried out classification for recognizing locomotion modes. Studies using a Machine Learning regression approach were excluded.The articles must be related to locomotion in various environments, e.g., level ground walking, stair ascent/descent, ramp ascent/descent, obstacle clearance, walking on a cross-slope, turning, walking on different surfaces, ... Studies were included if at least two locomotion modes were investigated.Only lower limb assistive devices such as exoskeletons, prostheses (for below or above knee amputation) or orthoses were considered.Studies were excluded if they met at least one of the following exclusion criteria: (1) non-human (robots or animals), (2) volunteers who are minors (under 18 years old), (3) studies focusing on volunteers equipped with an upper-limb device.

### 2.2. Information Sources

The PubMed and Web of Knowledge (including Web of Science core collection, Derwent Innovation Index, Russian Citation index, SciELO Citation Index) databases were searched on 31 July 2020. The two search strings used are given in the Appendix A. Published articles in English between 1 January 2000 and 31 July 2020 were included. Systematic reviews and meta-analyses were excluded. Conference papers were excluded if a corresponding published peer-reviewed article by the same authors had been included. Additional articles were included by further searching the references within the papers which were first identified by the search strategy described above.

### 2.3. Study Selection

The search strings were defined and validated by all authors. One person (FL) performed the initial search and removed the duplicates. Two main readers (DL, FL) independently screened the titles and the abstracts of all articles identified during the initial search. In case of disagreement, a third reader (LC) decided to include/exclude the article. Afterwards, the two readers (DL, FL) read the full text of the articles which had been picked from the previous step and checked them for eligibility using the criteria of our Modified QualSyst Tool which can be found in the Appendix A of this article. The process used to create the Modified QualSyst Tool can be found in the Section 2.4.1. Any disagreements on the eligibility of an article were resolved by the third person (LC).

### 2.4. Quality Assessment in Included Articles

The quality of the included articles was assessed with a dedicated QualSyst Tool [7] modified for the purposes of studying Machine Learning algorithms implemented on locomotion assistive devices. In the sections below, we provide further explanations of the score assignment for each article using this tool.

#### 2.4.1. Creating the Modified QualSyst Tool

Our first step was to remove irrelevant items from the QualSyst Tool [7] (Criteria 3 and 5 to 12, e.g., blinding of investigators, of subjects, etc.). Next, we added items which are relevant to the implementation of Machine Learning algorithms such as analysis windows, selected features, evaluation method of the algorithm, etc. All items were validated by all the authors and the quality of included articles was assessed by the main readers (FL, DL). The final version of this Modified QualSyst Tool can be found in the Appendix A of this article.

#### 2.4.2. Rating Articles Using the Modified QualSyst Tool

Twelve items were used for rating the articles. For each item, the article was rated with a score between 0 and 2 (with 2 indicating full supply of information, 1 a partial supply and 0 no information provided). Guidelines to allow consistent ratings across the included papers were created. These guidelines are provided in the Appendix A. The description of the twelve items were as follows:The first two items evaluated if the hypotheses and objectives of the study were sufficiently described and if the study design was appropriate.Item 3 evaluated if the volunteer characteristics were sufficiently described.Items 4 to 10 evaluated if the Machine Learning approach was sufficiently described to allow repeatability.Items 11 and 12 evaluated if the results were reported with enough details and if the conclusions were in accordance with them.

The score of each article was computed as the average of the 12 rated items. The maximum score possible for an article was 2. The score from 0–2 was transformed to a scale of 0–100% for ease of comprehension (0 indicating no information provided at all and 100 with maximum lucidity). More details on this scoring procedure and the guidelines used can be found in the Appendix A.

### 2.5. Synthesis of the Results

The following elements were extracted and grouped from the included studies:Investigated population (pathology and number of volunteers) and type of assistive device (above-knee prosthesis or below-knee prosthesis or orthosis or exoskeleton).The main elements of the experimental protocol are reported.
○The studied locomotor activities along with the walking speed of the volunteers are given.○The ‘Critical Timing’ is reported. It is the latest moment when the behavior of the locomotion assistive device can be adapted to the new locomotion mode without disturbing the user.○The type of sensors used in each study along with the total number of measurement axes per sensor are reported.○Details on the machine learning algorithm implementation are also reported (online and/or offline implementation; forward prediction and/or backward recognition [8]).The signal processing techniques and Machine Learning algorithms used are reported as well:
○This includes the type and length of the analysis windows.○The extracted features used for the analyses. If several configurations were tested, only the optimal configuration is given.○The machine learning algorithms are provided. Overall results of the machine learning algorithms are reported in terms of accuracy (A). So, if studies indicated the error rates (E), the corresponding mean overall accuracy was computed (A = 100—E in percent). For studies recruiting both healthy volunteers and patients, the reported accuracy of the machine learning algorithms corresponded to the patients (accuracy).

## 3. Results

### 3.1. Study Selection

The literature search produced 288 articles on PubMed and 1078 articles on Web of Science. Additionally, four studies were manually identified from references in the articles and added to the review. After removing the duplicates, there remained 1343 articles for screening. On the basis of titles and abstracts screening, 1267 articles were excluded from the review. Two authors independently read the full texts of the remaining 76 articles and checked them for eligibility. Finally, 58 articles were considered eligible to be included in this review. The PRISMA Flow Chart [6] is provided (Figure 1).

### 3.2. Quality of the Included Studies

The mean quality score of each study using the items of the Modified QualSyst Tool is provided in Table 1 and the detailed quality scores are presented in the Appendix A. The mean quality score was 68.4% +/− 13.4 for the articles.

### 3.3. Extracted Elements of the Included Studies

In this section, we summarize some of the key aspects of the extracted elements of the included studies.

#### 3.3.1. Type of Assistive Device and Related Population

The type of assistive device used in each study and the related population are detailed in Table 1.

Four types of devices were used in the included studies: prostheses for transfemoral amputation (i.e., above-knee prostheses), prostheses for transtibial amputation (i.e., below-knee prostheses), exoskeletons and orthoses.

**Above-knee prostheses.** This was the largest group among the published studies (N = 32). Among these thirty-two studies, the recruited population were either patients with unilateral transfemoral amputation or knee disarticulation (N = 19). There were healthy volunteers and patients with transfemoral amputation or knee disarticulation (N = 10). Finally, there were healthy volunteers wearing an above-knee prosthesis with an L-shape adaptor (N = 3).**Below-knee prosthesis.** This was the second largest group in this review (N = 18). Among those eighteen studies, the recruited population were either patients with unilateral transtibial amputation (N = 13) or healthy volunteers or patients with unilateral transtibial amputation (N = 5).**Exoskeletons** and **orthoses.** This constituted the smallest group in this review (N = 6 and N = 2 respectively). Among those eight studies, the recruited population was always healthy volunteers wearing the assistive device.

#### 3.3.2. Locomotion Activities and Walking Speed

The locomotion activities and walking speed investigated in each study are reported in Table 2.

The most representative experimental protocol investigated level ground walking along with stair and ramp ascent/descent activities (N = 43). Secondly, in some studies, level ground walking was investigated only with stair ascent and/or descent activities (N = 13). Among those fifty-six (43 + 13) studies, additional activities were also considered such as obstacle clearance (N = 6), turning (N = 2) or squatting (N = 1/58) for ‘dynamic’ activities and standing (N = 23/58) or sitting (N = 6/58) for static activities. The remaining two papers investigated level ground walking with cross slope walking (N = 1) and level ground walking with turning (N = 1).

In most studies, the walking speed was not provided (N = 33). One can assume that the volunteers walked at a self-selected speed in these thirty-three studies. Next, the volunteers were asked to walk at a self-selected speed in seventeen studies (N = 17). Finally, a small number of studies investigated different walking speeds: volunteers were asked to walk either at self-selected speed or at a slower or faster pace for different locomotion activities (N = 6). In the two remaining studies, recruited volunteers were asked to walk at a predefined speed of 0.7 m/s (N = 2).

#### 3.3.3. Identifying the Critical Timing

The Critical Timings used in each study are provided in Table 2.

Among the studies focusing on ankle-knee or ankle-foot prostheses (N = 50), most investigated the transitions between locomotion modes (N = 39). Several definitions of critical timing were used. We describe these definitions below:

Firstly, a study (N = 1) conducted by Huang et al. [23] in 2010 defined the critical timing as 200 ms before the prosthesis foot off of the ground for all transitions. Figure 2 illustrates the critical timing used in Huang et al. [23] for both level ground walking to stair ascent and stair descent to level ground walking transitions.

Secondly, some studies (N = 15) (in Huang et al. 2011 [24] for example) chose the critical timings at well-defined gait events (e.g., Foot-Off and Foot Contact): for transitions from level ground walking to any other locomotion mode, the critical timing was defined at the prosthesis foot off of the ground and for transitions from any locomotion mode to level ground walking, the critical timing was defined at prosthesis foot contact on level ground.

Thirdly, some studies (N = 5) (in Spanias et al. [42] for example) attempted to delay the critical timing in order to improve the locomotion mode prediction. Here, for transitions from level ground walking to any other locomotion mode, the critical timing was defined 90 ms after a gait event, such as the prosthesis foot off, mid-swing, prosthesis foot contact or mid-stance.

Finally, in a recent study (N = 1) conducted by Xu et al. [49] in 2018 defined the definitions of critical timings were altered based on the transition type and on the transitioning leg. As a result, the critical timing was delayed when the amputated leg was the leading leg for the transition. For level ground walking to stair ascent or stairs descent transitions, the critical timing was defined either at the last prosthesis foot off of the ground or at the first prosthesis foot contact on the stairs. For any other transitions, the critical timing was defined either at the first prosthesis foot contact on the new locomotion mode or at the first prosthesis foot off of the new locomotion mode.

The other studies did not investigate the transitions (N = 11) or did not report the critical timings used in the study (N = 17).

Among the studies focusing on orthoses or exoskeletons (N = 8), only three studies investigated the transitions between locomotion modes. In Long et al. [29], the critical timing occurred at foot contact of the contralateral leg of the exoskeleton. In Wang et al. [47], the critical timing occurred at foot contact of the ipsilateral leg in the new locomotion mode. Finally, in Zhou et al. [65], the critical timing occurred at mid-swing when the leg wearing the exoskeleton led the transition. It occurred at the last foot off of the ground for transitions from level ground to any other locomotion mode and finally for transitions from any locomotion mode to level ground walking it was at the first foot off of the ground. The remaining studies either did not investigate the transitions (N = 4) or did not report the critical timings used in the study (N = 1).

Panel A represents a patient with amputation in his transition from level walking to stair ascent. The superior part of this panel is a spatial representation of the patient motion. The line below is the temporal representation of the foot contact events. A dashed line maps the spatial representation to the temporal representation. For the spatial representation, the points refer to the spatial coordinates where the foot will hit/leave the ground. The temporal axis details the Foot Contact (FC) and Foot Off (FO) gait events for both sides. Critical timing is defined 200 ms prior to the prosthesis Foot Off event according to the Huang et al. Study [23]. The blue points are associated with the sound leg (Index S) and the red points are associated with the prosthesis side (index P). The panel B uses the same representation for patient from level walking to stair descent.

#### 3.3.4. Online/Offline Implementation of Machine Learning Algorithm for Prediction of the Upcoming Locomotion Mode or Recognition of the Current Locomotion Mode

Information regarding the type of implementation of the Machine Learning algorithms is provided in Table 2: recognition and/or prediction algorithm and online and/or offline implementation.

The Machine Learning algorithms developed in the studies included in this systematic review were designed either to predict the upcoming locomotion mode (N = 30) or to recognize the current locomotion mode (N = 24). Some studies developed a Locomotion Mode Recognition system with adaptive strategies (N = 4). A forward predictor identified the upcoming locomotion mode while a backward estimator recognized the current locomotion mode. The backward estimator was used to label new data and the forward predictor could be updated with these newly labeled data.

Most of algorithms were trained and evaluated offline (N = 40) with a few which were trained offline and evaluated online (N = 18).

#### 3.3.5. Data Type and Sensors Used

The details concerning the sensors used in the studies are provided in Table 2.

Sensors used in the included studies were of four types:Kinematic data were measured with sensors such as Inertial Motion Units (IMUs) (N = 36), or angle encoders (N = 21).Kinetic data such as interaction force between the device and the user were measured with load cell (N = 31). Ground reaction force was measured with foot insoles (N = 17) and torque at the joint was measured with motor current sensors (N = 14) or by measuring the length of a spring (N = 1).Physiological data were measured with sensors such as Electromyographs (EMG) (N = 21), Capacitive Sensing Systems (CSS) (N = 4) or Forcemyographs (FMG) (N = 1).Extrinsic data such as the distance between the user and an upcoming obstacle were measured with laser distance meters (N = 2) or with depth cameras (N = 2).

#### 3.3.6. Analysis Windows

The details concerning the analysis windows used in each study are provided in Table 3.

Three types of analysis windows can be distinguished: sliding (N = 30), unique (N = 7) or multiple (N = 19) windows. The first method consisted of using a sliding analysis window by defining a window length and a window increment. The windows can therefore overlap. For the unique and multiple methods, the analysis window(s) was (were) defined either by a starting or ending point and a fixed window length (N = 21) or by both end points with a variable window length (N = 5). The remaining studies (N = 2) did not provide any information concerning analysis windows.

#### 3.3.7. Features

The detailed features and domains used in each study can be found in Table 3.

Two main domains of features have been investigated in the included studies: time-domain (e.g., mean, minimum, maximum, standard deviation, etc.) (N = 48) and time-frequential domain features (e.g., coefficients of the wavelet transform) (N = 1). One study compared the performances of machine learning algorithm using either time-domain features or time-frequency domain features [9].

The remaining studies did not provide any information concerning the features used (N = 1) or used the temporal data measured by the sensors and did not extract any features (N = 7).

#### 3.3.8. Machine Learning Algorithms and Their Accuracies

Details on the machine learning algorithms used in studies and their reported accuracies are presented in Table 3.

Most of the studies used the classical pattern recognition algorithms (Bishop 2006 [66]) which are available. Three algorithms were implemented more often than others: Linear Discriminant Analysis (LDA) (N = 29), Support Vector Machines (SVM) (N = 17) and Dynamic Bayesian Network (DBN) (N = 10). Other algorithms were investigated a few times. Quadratic Discriminant Analysis (QDA) (N = 8) was used either with data from Capacitive Sensing Systems (N = 4) or from Inertial Motion Units (N = 4). Small Artificial Neural Networks (ANN) with 1 or 2 hidden layers were used to recognize the current locomotion mode (N = 6) or to predict the upcoming locomotion mode (N = 1). Convolutional Neural Networks (CNN) (N = 4) have started to be applied more recently for locomotion mode classification (since 2019). CNNs were essentially used to avoid feature selection: all studies using CNNs did not extract any feature and instead fed raw sensor data into the algorithm. Other algorithms were used only once (K-Nearest Neighbors—KNN and Long Short-Term Memory neural networks - LSTM) or twice (Decision Tree—DT and Hidden Markov Model—HMM).

Some less typical adaptive algorithms were also sometimes used. Learning From Testing data (LIFT) and Entropy Based Algorithm (EBA) were each used twice and Transductive SVM was used once [15,27].

## 4. Discussion

This systematic review included 58 articles implementing Machine Learning classifiers designed to identify the locomotion mode of assistive device user. Such algorithms were generally implemented as high-level controllers able to automatically adapt the behavior of lower limb prostheses, exoskeletons, or orthoses. We used the PubMed and Web of Science core collection databases for finding our references. This was done because most medical related literature (including biomedical engineering) can be found in these two databases. In addition, we performed an extensive search through the references of the papers from the aforementioned databases. As we were focusing on medical literature, we did not include Scopus as one of the databases for this review. This may have led to a very small number of papers that have not been included in this review.

Accuracy and the robustness (e.g., stable performance in the face of long-term use) of the algorithm were the variables most often used to report the results from studies investigating locomotion on different terrains. The influence of (1) sensors, (2) analysis windows and features, (3) machine learning algorithms on the accuracy and on the robustness of the locomotion mode classifiers are discussed below. It should be noted that the accuracies reported in this review are those which were supplied in each paper. Since each study was conducted with different circumstances such as number of subjects and conditions tested, accuracies can be compared within each study but cannot be compared between studies with precision.

### 4.1. Influence of Sensor Choice

Several sensors have been used to build locomotion mode classifiers. The choices of these sensors may influence the accuracy and the robustness of the classifiers. More details are provided in the sections below.

#### 4.1.1. Algorithm Accuracy

Among the included studies the three most used sensors were Inertial Motion Units (IMU) (N = 36, see Table 2), load cells (N = 31, Table 2) and electromyographs (EMG) (N = 21, Table 2).

Firstly, IMUs measure the acceleration and the rotational speed along three orthogonal axes. For example, Stolyarov et al. [43] classified level-ground walking (LW), stair ascent (SA), stair descent (SD), ramp ascent (RA) and ramp descent (RD) with LDA. They showed that including trajectory information of the prosthesis increased the averaged accuracy compared to using only the accelerations and rotational speeds (from 80.9% to 94.1%). They suggested using filtering techniques to reduce drift (e.g., Kalman filters, particle filters, etc.). These researchers also brought up the point that the performance of the classification algorithms might be reduced when applied to gait at slow walking speed. Other researchers demonstrating the capacity of IMUs for the detection of locomotion mode were Zhou et al. [65]. They were able with the SVM to classify three locomotion modes (LW, SA, SD) with the exclusive use of IMU data. They achieved above 90% accuracy using orientation information. The signals combining acceleration, rotational speed and orientation were directly extracted from the IMUs (MPU 9250, Ivensense^®^—the filter technique was not reported in the data sheet of the sensor).

However, these studies suggested that the algorithm performances could increase when fusing IMUs signals with other sensors signals. Thus, in most studies using IMUs, information from this sensor was fused with measurements from other sensors (see below).

Secondly, load cells measured the interaction force between the device and the user. For example, Huang et al. [24] classified five locomotion modes (LW, SA, SD, RA, RD) with LDA and SVM by using only a 6 degrees of freedom (DOF) load cell mounted on the prosthetic pylon of an above-knee prosthesis. The phase-dependent strategy achieved 85 to 95% accuracy during stance phase (Initial Double Limb Stance (DS1), Single Limb Stance (SS) and Terminal Double Limb Stance (DS2)) but the accuracies dropped to 50–60% during swing (SW) phase for both LDA and SVM classifiers. Similar drops in accuracy were reported when using only plantar pressure measurements [13,46]. According to the authors [24], the low classification accuracies in the swing phase were almost certainly due to low forces/moments generated during swing phase.

Thirdly, EMG signals measured from the residual limb were reported to contain useful information for locomotion mode predictions in early studies. Indeed, for example, Huang et al. [24] and Miller et al. [33] achieved classification of five locomotion modes (LW, SA, SD, RA, RD) using EMG signals measured in the residual limb of patients with transfemoral and transtibial unilateral amputation respectively. LDA and SVM classifiers were used in both studies. For volunteers with transfemoral amputation [24], the SVM achieved an accuracy of above 90% for all phases. The LDA algorithm achieved similar accuracies in the stance phase but a slightly lower accuracy of 85% in the swing phase. For volunteers with transtibial amputation [33], both LDA and SVM algorithms achieved around 98% accuracy. Many researchers have pointed out that the EMG signals suffer from disturbances especially because of shifts in electrode position when donning and doffing a prosthesis for example. Miller et al. [33] reported a mean loss in accuracy of 15.8% and 23.1% for LDA and SVM classifiers when the medial gastrocnemius electrode was shifted. Both studies concluded that EMG signals could be helpful for classifying locomotion modes as long as the signals are not disturbed. Several studies have provided suggestions for reducing these problems. They are discussed in the ‘Algorithm robustness’ Section 4.1.2 below.

Finally, sensor fusion has been proven to significantly increase accuracies of locomotion mode classifiers [24,54]. For example, Huang et al. [24] observed an increase in accuracy by combining EMG and load cell data instead of using either only EMG data or only load cell data (accuracy increase of up to 5.9% for an SVM classifier). Since then, data from different sensors have been fused together to reach higher accuracies. In another example, Young et al. [54] used 13 mechanical sensors (IMU, load cell, position, velocity and torque at knee and ankle joints) and recorded EMG signals from 9 muscles of the residual limb of volunteers with a transfemoral amputation. A DBN algorithm predicting upcoming locomotion modes reached 99% accuracy for steady-state steps and 88% accuracy for transitional steps.

#### 4.1.2. Algorithm Robustness

Sensors measurement noise over time can affect the performances of locomotion mode classifiers. To achieve reliable behavior of locomotion assistive device for long-term use, the influence of such noise should be considered. Techniques implemented to take into account sensors noise are discussed here.

EMG signals were mostly reported to be disturbed by environmental noise, electrode conductivity changes, shifts in electrode position or even loss of electrode contact [67,68]. Three techniques have been used to cope with such disturbances. The first one aims at training ML algorithm with several electrode displacement configurations [33]. The second one consists of building a sensor fault detection system so that disturbed EMG channel are removed if detected as noisy [23,40]. The third one uses an adaptive framework so that ML algorithm can be updated when EMG signals are disturbed [42]. The latter adaptive algorithm also included a sensor fault detection system. Alternatively, according to some researchers [62,63], capacitive sensing systems, measuring the gap change between the residual limb and the prosthetic socket [63], could eventually replace EMG signals since such sensors appear to be robust to donning and doffing an ankle-knee prosthesis and to load bearing changes [62].

### 4.2. Influence of Analysis Windows

In this section, we will discuss the influence of the analysis window configuration on the accuracy of locomotion mode prediction.

Among the included studies, sliding (N = 30, Table 3) and multiple (N = 19, Table 3) analysis windows were the preferred configurations. While the implementation of sliding windows requires the building of one classifier per gait phase, the implementation of multiple windows is performed by building one classifier per analysis window [50,51,52,53,54,55]. The number of classifiers depends on the number of gait phases for sliding windows and depends on the number of windows for multiple windows. In the case of sliding windows, Chen et al. [12] observed that the number of phases for phase-dependent classification significantly influences algorithm accuracy. As a result, using four gait phases (DS1, SS, DS2, SW) increased the accuracy of both LDA and QDA compared to when using only two phases (Stance, Swing). As the sliding window method generally involves a longer portion of the gait phase in question, the data to be classified are generally more variable.

Several studies reported that the length of analysis windows had a significant impact on algorithm performances for multiple window [53,54] and sliding window [22] configurations. Young et al. [53,54], using multiple windows, showed that there was an optimal window length (between 200 and 300 ms) for classification accuracies using mechanical and EMG data for both steady state and transitional data. The same was found in a study using sliding windows, where the length of the window but not its increments were found to affect algorithm performances [22]. For online implementation however smaller window increment ensures a faster response time since locomotion mode classification is performed more often.

More recently, some researchers did not use analysis windows which ended at classic gait events like foot contact but instead allowed for a delay in the termination of the analysis window. For example, Simon et al. [37,69] had an analysis window which ended 90ms after foot contact or foot off. This delay increased the accuracy of a DBN algorithm and did not affect the stability of the users of a powered above-knee prosthesis.

### 4.3. Influence of Features

The features set used in each study was highly dependent on the sensors used.

For EMG signals, two types of features were tested: (1) time-domain features and (2) time-frequential domain features. The most commonly used time-domain features were mean absolute value, waveform length, number of zero crossings, number of slope sign changes (N = 21) and the coefficients of autoregressive models (N = 8). For time-frequential domain features, the coefficients of the wavelet transform of EMG signals were used once [9]. Ai et al. [9] compared LDA and SVM performances when using time-domain features or time-frequency domain features. Both algorithms reached higher accuracies with time domain features for one volunteer with below-knee amputation, e.g., in the case of the SVM 91.9% with time-domain features vs. 82.3% with time-frequency features. Additionally, time-domain features were easier and faster to compute [9].

A large number of studies (N = 48, Table 3) used mechanical sensors (IMU, load cells, encoders, pressure insoles, etc.). The most representative feature (N = 34, Table 3) set was a combination of the following time-domain features: mean, maximum, minimum and standard deviation. Initial and final values were also sometimes added to the feature set (N = 8, Table 3).

Finally, several feature reduction techniques were sometimes used to find the minimal feature set necessary for successful classification and to avoid overfitting (N = 14): Wrapper techniques such as Sequential Forward Selection (SFS) and Selection Backward Selection (SBS) were used to pick the features having the highest impact on the classification accuracy [39] (N = 8). Such methods are time consuming [18]. Zhang et al. [57] compared the processing time taken by two wrapper methods and a filter method. The filter method was found to be faster compared to wrapper methods (84 s for the filter method vs. 1978 s for SBS).

### 4.4. Influence of Machine Learning Algorithm

#### 4.4.1. On Accuracy

A variety of ML algorithms were used in the included studies. The most frequently used algorithms were LDA (N = 29, Table 3), SVM (N = 19, Table 3) and DBN (N = 10, Table 3). Also, CNNs were used to avoid features selection (N = 4, Table 3).

LDA is easy to implement since no hyperparameters need to be tuned [48,70]. This algorithm is fast (1.29 ms [48], 0.078 ms with parallelization [32]) and not prone to overfitting [9]. For these reasons, this algorithm is often used as a baseline for performance comparisons between several algorithms [32,42]. More importantly, in some studies, LDA obtained accuracies similar to neural networks [48] and to SVM [33].

Even though, hyperparameters such as kernel parameter and the penalty factor need to be tuned for SVM [16], optimization techniques (e.g., grid search [9], particle swarm optimization [29]) have been found in some studies to reach slightly better performances than LDA [9,24] or QDA [62].

One of the first researchers to use DBNs were Young et al. in 2013 [50,51]. By adding past information to those of the current state, the DBN was able to obtain higher classification accuracies than LDA [54] (88% vs. 85% for transitional accuracies for DBN and LDA respectively). The DBN, unlike LDA with uniform priors, take transitional probabilities into account (e.g., in stair ascent mode, the next mode is more likely to be stair ascent or level ground walking).

Finally, CNNs were recently used in a few studies [16,44,58,59]. For example, Zhang et al. [58,59] used depth-images with a depth-camera coupled with an IMU mounted on the prosthetic pylon of an above-knee prosthesis. CNNs, known to perform well when handling image datasets are often used to avoid manual feature selection. CNNs were also used in the case of non-image data, e.g., IMU data [44] or load cell data [16]. All four studies using CNNs reported an accuracy above 89% but none of those studies implemented the designed CNN online.

The most common mistake was misclassification between ramp ascent and level ground walking modes [50]. Grouping ramp ascent and level walking classes were reported to improve the performances of locomotion mode classifiers [50]. Such a technique is relevant when the control laws (impedance in [43,50]) are similar for both modes. Zhang et al. [59] evaluated the influence of such errors (misclassifications between level walking and incline walking) on the stability of the user of an above-knee prosthesis using angular momentum and a subjective questionnaire. It was observed that the effect of the errors depends on the type of error, the error duration, and the gait phase where the error occurred. Errors were considered critical if the stability of prosthesis users was disturbed. This appears to be a good criterion for evaluating the importance of errors when designing a locomotion mode classifier.

#### 4.4.2. On Robustness

Very few studies have evaluated the performances of locomotion mode classifiers for long term use. Adaptive frameworks have been proposed to deal with EMG disturbances [42] or to achieve stable performances for long term use [27]. For example, Spanias et al. [42], designed a forward predictor and a backward estimator. The forward predictor is an ML algorithm designed to predict the upcoming locomotion mode of an assistive device user. The backward estimator is an ML algorithm designed to recognize the current locomotion mode. The latter algorithm was used to label new data. Then, the newly labelled data were incorporated into the training set and then used to update the forward predictor parameters. Spanias et al. [42] used this framework to deal with EMG disturbances. The adaptive algorithm learned to reincorporate disturbed EMG channels over time. The adaptive algorithm was reported to perform significantly better than a non-adaptive algorithm. In another example, Liu et al. [27] evaluated the performance of adaptive algorithms compared to a non-adaptive algorithm across multiple session within a single experimental day. After donning and doffing the prosthesis, the adaptive algorithms were reported to update classifiers boundaries and to recover initial accuracy whereas the performances of the non-adaptive algorithms gradually decreased. To sum up, adaptive frameworks seem to be a promising solution to achieve long-term locomotion mode classification.

### 4.5. Propositions for Future Work

This systematic review included 58 articles published between 1 January 2000 and 31 July 2020. All 58 articles implemented ML-based locomotion mode classifiers designed for users of lower limb assistive devices. As can be seen from Table 3, classification accuracies under the tested conditions were almost always very high, hence indicating good progress in the attempts to construct more intelligent prosthetic devices. Nevertheless, there is always room for improvement. We try here to propose some recommendations concerning the research reports in the field as well as suggestions for moving forward with the implementation of these devices in the daily lives of the patients.

#### 4.5.1. Homogenization of Reports

We will start first with the question of terms that are used in the field. This is not a trivial matter as the homogenization of terms would increase the understanding between researchers and hence speed up progress. There is much confusion around the use of terms recognition and prediction. The two terms are used in an interchangeable manner across studies but do not refer to the same goal. We propose that classifying the locomotion mode before the critical timing can be considered as a prediction task while a classification made after the critical timing can be considered as a recognition task. For recall, the critical timing is the latest moment when the behavior of the locomotion assistive device can be adapted to the new locomotion mode without disturbing the user. A more discriminating use of the two terms, recognition and prediction, would ease the comprehension of the studies.

The report of accuracies also suffers from a similar lack of precision. While many reports have distinguished between accuracies during steady state and the transitional step, several have not. Adding together the success obtained in steady state with that which is obtained in transitional steps is misleading, as the errors made in the latter tend to be higher. We therefore propose that there should be a systematic distinction of accuracies for these two modes.

#### 4.5.2. Recommendations for Generalization to Daily Life Conditions

The review shows that significant progress has been made in the efforts to ease the use of prosthetic devices across multiple terrains. Nevertheless, some obvious steps are necessary to move ahead with ensuring the comfortable use of these devices in the daily lives of the patients.

An obvious thing to add on the list would be the inclusion of more daily life conditions for testing the devices. Examples of these would be different angles of approaches towards stairs or slope [9,28], different staircases [18] or load bearing changes [51]. A good extension for many of the studies included in this review would be a test of the algorithms outside the laboratory. Only a very small number of studies managed to take this step. For example, the work of Zhang et al. [58,59] evaluated a CNN classifier with data acquired both indoors and outdoors. Such studies are to be encouraged.

Another important condition to be included, to make the prosthetic devices more usable in daily life, would be the integration of multiple speeds in the study. Once again, very few researchers have investigated this condition. One researcher who has taken a step in this direction is Liu et al., 2017 [28].

A third variation which is not often taken into consideration is the transitioning leg which is used when entering a new terrain. While subjects tend to use one leg more than the other when crossing into new conditions, the side used is not always identical and subjects can change the transitioning leg. A handful of studies such as one by Zhou et al. [65] have taken this into account. They reported better accuracies when the locomotion mode classifier was trained with data from both transitioning legs. This may be a simple condition to include in more of the future studies in the field.

We turn here, from a discussion of conditions to be tested, to comments on how to decrease the burden of developing an algorithm which is tuned to each patient. The process of gathering data for the purpose of training the ML algorithm for each patient can be long and burdensome. A few researchers have provided recommendations on how to reduce the difficulty of this step. For example, Zhang et al. [60] proposed an automatic training method through environmental sensing. A radar distance meter coupled with an IMU helped to sense the environment and to automatically label the acquired data. Automatic labelling could also be achieved with depth cameras [58,59]. Another step in this direction has been the use of subject independent models which could potentially reduce the amount of training data needed. Efforts of this type have been made by Young et al. [52] and Spanias [8]. It seems that the addition of this step to future investigations of predictive or recognition algorithms would provide the additional bonus of reduced training time for the patient.

## Figures and Tables

**Figure 1 sensors-20-06345-f001:**
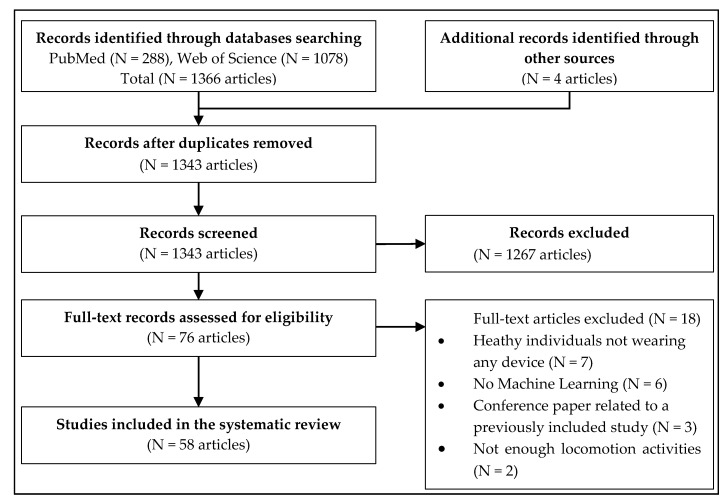
Preferred Reporting Items for Systematic reviews and Meta-Analyses (PRISMA) flow chart of the systematic review.

**Figure 2 sensors-20-06345-f002:**
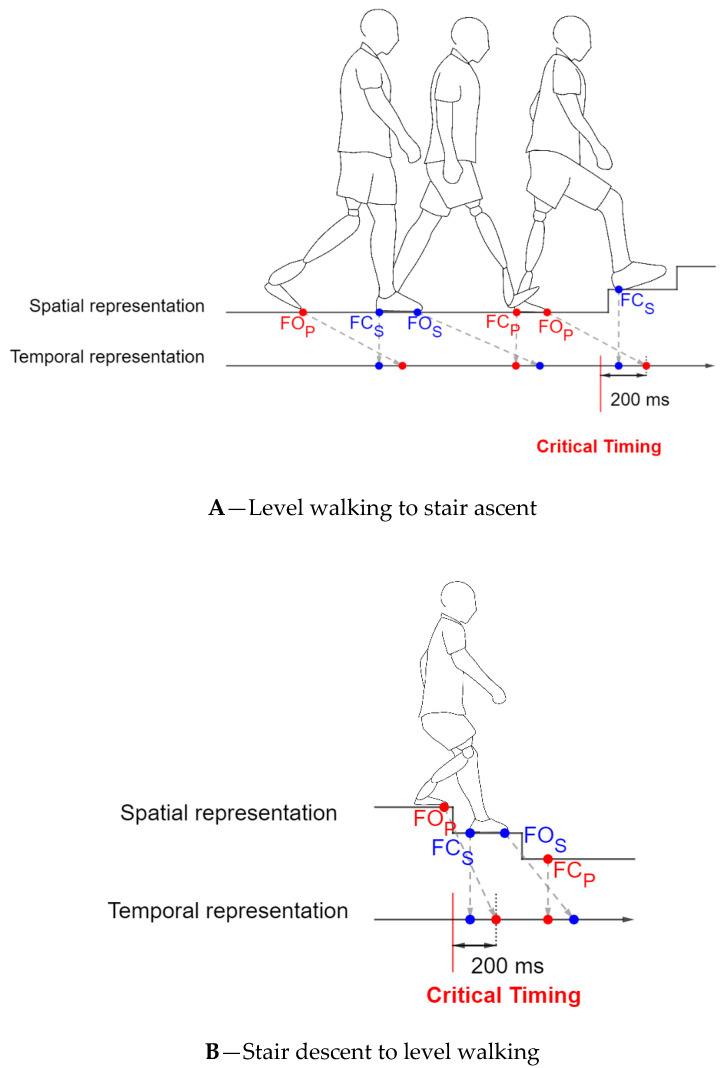
Example of the critical timings used in Huang et al. [23].

**Table 1 sensors-20-06345-t001:** Quality assessment and recruited volunteers in the included studies.

Article	Quality Score	Groups (N)	Locomotion Assistive Device
Ai et al. 2017 [9]	70.8%	TT (4)/Healthy (1)	Ankle Prosthesis
Beil et al. 2018 [10]	90.9%	Healthy (10)	Exoskeleton
Chen et al. 2013 [11]	72.7%	TT (5)/Healthy (8)	Ankle Prosthesis
Chen et al. 2014 [12]	79.2%	TT (1)/Healthy (7)	Ankle Prosthesis
Chen et al. 2015 [13]	77.3%	TT (1)/Healthy (5)	Ankle Prosthesis
Du et al. 2012 [14]	75.0%	TF (9)	Ankle Knee Prosthesis
Du et al. 2013 [15]	45.8%	TF (4)	Ankle Knee Prosthesis
Feng et al. 2019 [16]	77.3%	TT (3)	Ankle Prosthesis
Godiyal et al. 2018 [17]	86.4%	TF (2)/Healthy (8)	Ankle Knee Prosthesis
Gong et al. 2018 [18]	86.4%	Healthy (1)	Orthosis
Gong et al. 2020 [19]	86.4%	Healthy (3)	Orthosis
Hernandez et al. 2012 [20]	37.5%	TF (1)	Ankle Knee Prosthesis
Hernandez et al. 2013 [21]	54.2%	Healthy (1)	Ankle Knee Prosthesis
Huang et al. 2009 [22]	81.8%	TF (2)/Healthy (8)	Ankle Knee Prosthesis
Huang et al. 2010 [23]	79.2%	TF (1)/Healthy (5)	Ankle Knee Prosthesis
Huang et al. 2011 [24]	83.3%	TF (5)	Ankle Knee Prosthesis
Kim et al. 2017 [25]	63.6%	Healthy (8)	Exoskeleton
Liu et al. 2016 [26]	70.8%	TF (1)/Healthy (6)	Ankle Knee Prosthesis
Liu et al. 2017 [27]	66.7%	TF (2)/Healthy (2)	Ankle Knee Prosthesis
Liu et al. 2017 [28]	63.6%	TF (2)/Healthy (3)	Ankle Knee Prosthesis
Long et al. 2016 [29]	83.3%	Healthy (3)	Exoskeleton
Mai et al. 2011 [30]	50.0%	TT (1)	Ankle Prosthesis
Mai et al. 2018a [31]	45.8%	TT (1)	Ankle Prosthesis
Mai et al. 2018b [32]	54.2%	TT (1)	Ankle Prosthesis
Miller et al. 2013 [33]	90.9%	TT (5)/Healthy (5)	Ankle Prosthesis
Moon et al. 2019 [34]	33.3%	Healthy (1)	Exoskeleton
Pew et al. 2017 [35]	66.7%	TT (5)	Ankle Prosthesis
Shell et al. 2018 [36]	70.8%	TT (3)	Ankle Prosthesis
Simon et al. 2017 [37]	66.7%	TF (6)	Ankle Knee Prosthesis
Spanias et al. 2014 [38]	54.2%	TF (4)	Ankle Knee Prosthesis
Spanias et al. 2015 [39]	54.2%	TF (6)	Ankle Knee Prosthesis
Spanias et al. 2016a [40]	62.5%	TF (8)	Ankle Knee Prosthesis
Spanias et al. 2016b [8]	58.3%	Healthy (2)	Ankle Knee Prosthesis
Spanias et al. 2017 [41]	58.3%	TF (3)	Ankle Knee Prosthesis
Spanias et al. 2018 [42]	62.5%	TF (8)	Ankle Knee Prosthesis
Stolyarov et al. 2017 [43]	79.2%	TF (6)	Ankle Knee Prosthesis
Su et al. 2019 [44]	77.3%	TF (1)/Healthy (10)	Ankle Knee Prosthesis
Tkach et al. 2013 [45]	62.5%	TT (5)	Ankle Prosthesis
Wang et al. 2013 [46]	66.7%	TT (1)	Ankle Prosthesis
Wang et al. 2018 [47]	79.2%	Healthy (22)	Exoskeleton
Woodward et al. 2016 [48]	91.7%	TF (6)	Ankle Knee Prosthesis
Xu et al. 2018 [49]	75.0%	TT (3)	Ankle Prosthesis
Young et al. 2013a [50]	66.7%	TF (4)	Ankle Knee Prosthesis
Young et al. 2013b [51]	79.2%	TF (6)	Ankle Knee Prosthesis
Young et al. 2013c [52]	62.5%	TF (4)	Ankle Knee Prosthesis
Young et al. 2014a [53]	66.7%	TF (6)	Ankle Knee Prosthesis
Young et al. 2014b [54]	75.0%	TF (8)	Ankle Knee Prosthesis
Young et al. 2016 [55]	75.0%	TF (8)	Ankle Knee Prosthesis
Zhang et al. 2011 [56]	70.8%	TF (1)/Healthy (1)	Ankle Knee Prosthesis
Zhang et al. 2013 [57]	66.7%	TF (4)	Ankle Knee Prosthesis
Zhang et al. 2019 [58]	63.6%	TF (3)/Healthy (6)	Ankle Knee Prosthesis
Zhang et al. 2019 [59]	59.1%	TF (3)/Healthy (6)	Ankle Knee Prosthesis
Zhang et al. 2012 [60]	62.5%	Healthy (1)	Ankle Knee Prosthesis
Zheng et al. 2013 [61]	86.4%	TT (1)	Ankle Prosthesis
Zheng et al. 2014 [62]	86.4%	TT (6)	Ankle Prosthesis
Zheng et al. 2016 [63]	75.0%	TT (6)	Ankle Prosthesis
Zheng et al. 2019 [64]	54.2%	TT (6)	Ankle Prosthesis
Zhou et al. 2019 [65]	54.2%	Healthy (3)	Exoskeleton

TT = Volunteer with a unilateral transtibial amputation, TF = Volunteer with a unilateral transfemoral amputation, N = Number of recruited volunteers.

**Table 2 sensors-20-06345-t002:** Sensors used for recognition and/or prediction of the locomotion activities investigated in the included studies.

Article	Locomotion Activities	Critical Timing	Speed	Sensors	Axes × Sensors	Offline/Online	Recognition/Prediction
Ai et al. 2017 [9]	LW, SA, SD, ST, SQ	NP	NP	EMG IMU	1 × 4 3(A) × 1	Off	R
Beil et al. 2018 [10]	LW, SA, SD, Turns, ST	NA	SSS	Force Sensors IMU	3 × 7 6(A, α) × 3	Off	R
Chen et al. 2013 [11]	LW, SA, SD, OBS, ST, SIT	NA	NP	Capacitive Pressure	1 × 10 2 × 1	Off	R
Chen et al. 2014 [12]	LW, SA, SD, RA, RD	3	NP	IMU Pressure	9(A, G, α) × 2 4 × 2	Off	R
Chen et al. 2015 [13]	LW, SA, SD, OBS, ST, SIT	NA	SSS	Pressure	4 × 1	Off	R
Du et al. 2012 [14]	LW, SA, SD, RA, RD	2	NP	EMG Load cell	1 × 9 6 × 1	Off	P
Du et al. 2013 [15]	LW, SA, SD, RA, RD	NP	NP	EMG Load cell	1 × 7 6 × 1	Off	P
Feng et al. 2019 [16]	LW, SA, SD, RA, RD	NA	NP	Load cell Angle Sensor	NP 1 × 1	Off	R
Godiyal et al. 2018 [17]	LW, SA, SD, RA, RD	NA	SSS	FMG Pressure	1 × 8 3 × 1	Off	R
Gong et al. 2018 [18]	LW, SA, SD, RA, RD, ST	NA	Imposed Speed	IMU	9(A, G, α) × 2	Off and On	R
Gong et al. 2020 [19]	LW, SA, SD, RA, RD, ST	NA	Imposed Speed	IMU	9(A, G, α) × 2	Off and On	R
Hernandez et al. 2012 [20]	LW, SA, SD, RA, RD, ST, SIT	NP	NP	Load cell EMG	6 × 1 1 × 7	Off	R
Hernandez et al. 2013 [21]	LW, SA, ST	2	NP	Load cell EMG	6 × 1 1 × 7	Off and On	P
Huang et al. 2009 [22]	LW, SA, SD, OBS, Turns, ST	NA	SSS	EMG Pressure	1 × 11 2 × 1	Off	R
Huang et al. 2010 [23]	LW, SA, SD, OBS	1	SSS	EMG Pressure	1 × 11 NP	Off	P
Huang et al. 2011 [24]	LW, SA, SD, RA, RD, OBS	2	SSS	EMG Load cell Pressure	1 × 11 6 × 1 NP	Off	P
Kim et al. 2017 [25]	LW, SA, SD, RA, RD	NA	NP	Joint angle IMU Load cell Pressure	1 × 4 3(α) × 5 1 × 4 4 × 1	Off	R
Liu et al. 2016 [26]	LW, SA, SD, RA, RD	2	SSS	EMG Load cell IMU Laser	1 × 8 6 × 1 6(A, G, α) × 1 1 × 1	Off and On	P
Liu et al. 2017 [27]	LW, SA, SD, RA, RD	NP	NP	EMG Load cell	1 × 7 6 × 1	Off and On	P
Liu et al. 2017 [28]	LW, SA, SD, RA, RD	NA	SSS, SL, F	IMU Pressure	4(A, G) × 1 2 × 2	Off	R
Long et al. 2016 [29]	LW, SA, SD, RA, RD	5	SSS	IMU Pressure	3(α) × 4 3 × 2	Off and On	P
Mai et al. 2011 [30]	LW, SA, SD	NA	SSS, F	Load cell	1 × 12	Off	R
Mai et al. 2018a [31]	LW, SA, SD, RA, RD, ST	NP	NP	IMU	9(A, G, α) × 2	Off and On	R
Mai et al. 2018b [32]	LW, SA, SD, RA, RD	NP	NP	IMU	8(3A, 3G, 2α) × 2	Off and On	R
Miller et al. 2013 [33]	LW, SA, SD, RA, RD	NA	SSS, (SL, F) for LW	EMG Pressure	1 × 4 2 × 2	Off	R
Moon et al. 2019 [34]	LW, SA, SD	NP	NP	Motor Encoder Spring length	1 × 1 1 × 1	Off and On	R
Pew et al. 2017 [35]	LW, Turns	NP	SSS	Load cell	6 × 1	Off	P
Shell et al. 2018 [36]	LW, cross-slope	NP	NP	IMU	5(3A, 2G) × 1	Off	R
Simon et al. 2017 [37]	LW, SA, SD, RA, RD, ST	3	SSS, (SL, F) for LW	Joint Angle Joint Velocity Motor Current IMU Load cell	1 × 2 1 × 2 1 × 2 6(A, G, α) × 1 6 × 1	Off	P
Spanias et al. 2014 [38]	LW, SA, SD, RA, RD	2	NP	Joint Angle Joint Velocity Motor Current IMU Load cell EMG	1 × 2 1 × 2 1 × 2 6(A, G, α) × 1 1 × 1 1 × 9	Off	P and R
Spanias et al. 2015 [39]	LW, SA, SD, RA, RD	2	SSS, SL, F	Joint Angle Joint Velocity Motor Current IMU Load cell EMG	1 × 2 1 × 2 1 × 2 8(3A, 3G, 2α) × 1 6 × 1 1 × 4	Off	P
Spanias et al. 2016a [40]	LW, SA, SD, RA, RD	2	NP	Joint Angle Joint Velocity Motor Current IMU Load cell EMG	1 × 2 1 × 2 1 × 2 6(A, G, α) × 1 1 × 1 1 × 9	Off	P
Spanias et al. 2016b [8]	LW, SA, SD, RA, RD, ST	3	NP	Joint Angle Joint Velocity Motor Current IMU Load cell	1 × 2 1 × 2 1 × 2 10(3A, 3G, 4α) × 1 6 × 1	Off and On	P and R
Spanias et al. 2017 [41]	LW, SA, SD, RA, RD, ST	3	NP	Joint Angle Joint Velocity Motor Current IMU Load cell	1 × 2 1 × 2 1 × 2 10(3A, 3G, 4α) × 1 6 × 1	Off and On	P and R
Spanias et al. 2018 [42]	LW, SA, SD, RA, RD, ST	3	NP	Joint Angle Joint Velocity Motor Current IMU Load cell EMG	1 × 2 1 × 2 1 × 2 10(3A, 3G, 4α) × 1 6 × 1 1 × 8	Off and On	P and R
Stolyarov et al. 2017 [43]	LW, SA, SD, RA, RD	NP	SSS	IMU	6(A, G) × 1	Off	P
Su et al. 2019 [44]	LW, SA, SD, RA, RD	NP	SSS	IMU	6(A, G) × 3	Off	R
Tkach et al. 2013 [45]	LW, SA, SD, RA, RD	NP	SSS	IMU Joint Angle Joint Velocity Joint Current EMG	3(A) × 1 1 × 1 1 × 1 1 × 1 1 × 4	Off	R
Wang et al. 2013 [46]	LW, SA, SD, ST, SIT	NA	NP	Pressure	4 × 1	Off	R
Wang et al. 2018 [47]	LW, SA, SD, ST, SIT	2	NP	Joint Angles	1 × 6	Off and On	P
Woodward et al. 2016 [48]	LW, SA, SD, RA, RD	NP	NP	IMU Joint Angle Joint Velocity Joint Current Load cell	7(3A, 3G, 1α) × 1 1 × 2 1 × 2 1 × 2 6 × 1	Off	P
Xu et al. 2018 [49]	LW, SA, SD, RA, RD, ST	4	NP	IMU Load cell	9(A, G, α) × 1 1 × 1	Off and On	P
Young et al. 2013a [50]	LW, SA, SD, RA, RD	2	NP	IMU Joint Angle Joint Velocity Joint Current Load cell EMG	6(A, G) × 1 1 × 2 1 × 2 1 × 2 1 × 1 1 × 9	Off	P
Young et al. 2013b [51]	LW, SA, SD, RA, RD	2	NP	IMU Joint Angle Joint Velocity Joint Current Load cell	6(A, G) × 1 1 × 2 1 × 2 1 × 2 1 × 1	Off	P
Young et al. 2013c [52]	LW, SA, SD, RA, RD	2	NP	IMU Joint Angle Joint Velocity Joint Current Load cell EMG	6(A, G) × 1 1 × 2 1 × 2 1 × 2 1 × 1 1 × 7	Off	P
Young et al. 2014a [53]	LW, SA, SD, RA, RD	2	SSS	IMU Joint Angle Joint Velocity Joint Current Load cell	6(A, G) × 1 1 × 2 1 × 2 1 × 2 1 × 1	Off	P
Young et al. 2014b [54]	LW, SA, SD, RA, RD	2	SSS, (SL, F) for LW	IMU Joint Angle Joint Velocity Joint Current Load cell EMG	6(A, G) × 1 1 × 2 1 × 2 1 × 2 1 × 1 1 × 9	Off	P
Young et al. 2016 [55]	LW, SA, SD, RA, RD	2	SSS	IMU Joint Angle Joint Velocity Joint Current Load cell	6(A, G) × 1 1 × 2 1 × 2 1 × 2 1 × 1	Off	P
Zhang et al. 2011 [56]	LW, SA, SD, RA, RD	2	NP	Load cell EMG	6 × 1 1 × 11	Off and On	P
Zhang et al. 2013 [57]	LW, SA, SD, RA, RD, ST, SIT	NP	NP	IMU Load cell EMG	6(A, G) × 2 6 × 1 1 × 8	Off and On	P
Zhang et al. 2019 [58]	LW, SA, SD, RA, RD	NP	NP	Depth Camera IMU	224 × 171 3(α) × 1	Off	P
Zhang et al. 2019 [59]	LW, SA, SD, RA, RD	NP	NP	Depth Camera IMU	224 × 171 3(α) × 1	Off	P
Zhang et al. 2012 [60]	LW, SA, SD, ST	2	NP	IMU Laser Load cell EMG	6(A, G) × 1 1 × 1 6 × 1 1 × 7	Off and On	P
Zheng et al. 2013 [61]	LW, SA, SD, RA, RD, OBS, ST	NA	SSS	Capacitive Pressure	1 × 7 3 × 1	Off	R
Zheng et al. 2014 [62]	LW, SA, SD, RA, RD, ST	NA	SSS	Capacitive Pressure	1 × 6 3 × 1	Off	R
Zheng et al. 2016 [63]	LW, SA, SD, RA, RD, ST	NP	SSS	IMU Load cell Joint angle Pressure Capacitive	8(3A, 3G, 2α) × 2 1 × 1 1 × 1 4 × 1 1 × 6	Off	P
Zheng et al. 2019 [64]	LW, SA, SD, RA, RD, ST	NP	NP	IMU Load cell Joint angle Pressure	8(3A, 3G, 2α) × 2 1 × 1 1 × 1 4 × 1	Off	P
Zhou et al. 2019 [65]	LW, SA, SD	3	NP	IMU Load cell Joint angle	9(A, G, α) × 2 1 × 2 1 × 1	Off and On	P

Locomotion Activities: LW = Level-ground Walking, SA = Stairs Ascent, SD = Stairs Descent, RA = Ramp Ascent, RD = Ramp Descent, ST = Standing, SIT = Sitting, SQ = Squatting, OBS = Obstacle clearance. Critical Timing: NP = Not Provided, NA = Not Applicable, 1 = 200 ms before the prosthesis foot off of the ground for all transitions, 2 = for transitions from level ground walking to any other locomotion mode, the critical timing was defined either at the prosthesis foot off of the ground or at mid-swing and for transitions from any locomotion mode to level ground walking, the critical timing was defined at prosthesis foot contact on level ground walking or at mid-stance, 3 = at foot off of the previous locomotion mode for all transitions,4 = For level ground walking to stairs ascent or stairs descent transitions, the critical timing was defined either at the prosthesis foot off of the ground or at prosthesis foot contact on the stairs. For any other transitions, the critical timing was defined either at the prosthesis foot contact on the new locomotion mode or at the first prosthesis foot off of the new locomotion mode,5 = the critical timing occurred at foot contact of the contralateral leg of the exoskeleton. Speed: NP = Not Provided, SSS = Self-Selected Speed, SL = Slow, F = Fast, (SL, F) for LW = Slower and faster paces tested only for level-ground walking. Sensors: EMG = ElectroMyoGraphs, IMU = Inertial Motion Unit, FMG = Force MyoGraph. Axes × Sensors: The number of measurement axes and the number of sensors is reported. For IMUs, the signals used are specified with A = Accelerometer, G = Gyroscope, α = Inclination. For instance, 9(A, G, α) × 2 means that 2 IMUs were used and for each IMU the 3D accelerations, the 3D rotational speed and the 3D orientation were extracted. Offline/Online: Off = Offline, On = Online. Recognition/Prediction: R = Recognition, P = Prediction.

**Table 3 sensors-20-06345-t003:** Preprocessing techniques, Machine Learning algorithms and reported accuracies of the included studies. The details of the preprocessing techniques (windows and features) and the machine learning algorithms used in each study are reported along with the corresponding accuracy. If several configurations were tested, only the optimal configuration is reported.

Article	Analysis Windows			Sensors	Features	Algorithm	Accuracy
	Type	Number	Length				
Ai et al. 2017 [9]	Sliding	NA	250	EMG Mech	WT DTW	SVM	97.9
Beil et al. 2018 [10]	Sliding	NA	300	Mech	Raw data	HMM	92.8
Chen et al. 2013 [11]	Sliding	NA	150	Capacitive	Mean, Max, Min, RMS	LDA	94.54
Chen et al. 2014 [12]	Sliding	NA	160	Pressure Mech	Mean, Max, Min, SD, RMS, WL, CORR Mean, Max, Min, SD, RMS, WL, ZC, CORR	LR	98.2
Chen et al. 2015 [13]	Multiple	4	200	Pressure	SD, AR	LDA	98.4
Du et al. 2012 [14]	Sliding	NA	150	EMG Mech	MAV, SSC, WL, ZC Mean, Max, Min	LDA	98
Du et al. 2013 [15]	Sliding	NA	160	EMG Mech	MAV, SSC, WL, ZC Mean, Max, Min	EBA	92.5
Feng et al. 2019 [16]	Unique	1	Gait Cycle	Mech	Raw Data	CNN	92.1
Godiyal et al. 2018 [17]	Unique	1	Stance	FMG	Mean, Max, Min, SD, RMS, WL, SSC, MAD	LDA	96.1
Gong et al. 2018 [18]	Sliding	NA	250	Mech	Mean, Max, Min, SD, MAD	ANN	97.8
Gong et al. 2020 [19]	Sliding	NA	250	Mech	Mean, Max, Min, SD, MAD	ANN	98.4
Hernandez et al. 2012 [20]	Sliding	NA	150	EMG Mech	MAV, SSC, WL, ZC Mean, Max, Min	SVM	NP
Hernandez et al. 2013 [21]	Sliding	NA	160	EMG Mech	MAV, SSC, WL, ZC Mean, Max, Min	SVM	99.9
Huang et al. 2009 [22]	Sliding	NA	140	EMG	MAV, SSC, WL, ZC	LDA	95.5
Huang et al. 2010 [23]	Multiple	3	100	EMG	MAV, SSC, WL, ZC	LDA	NR
Huang et al. 2011 [24]	Sliding	NA	150	EMG Mech	MAV, SSC, WL, ZC Mean, Max, Min	SVM	100
Kim et al. 2017 [25]	Unique	1	FC contro to FC ipsi	Mech	Custom values	DT	99.1
Liu et al. 2016 [26]	Sliding	NA	50	EMG Mech	MAV, SSC, WL, ZC Mean, Max, Min, SD	LDA	~98
Liu et al. 2017 [27]	Sliding	NA	160	EMG Mech	MAV, SSC, WL, ZC Mean, Max, Min, SD	EBA/LIFT	94.3
Liu et al. 2017 [28]	Unique	1	800	Mech	ICC	HMM	95.8
Long et al. 2016 [29]	NP	NP	NP	Mech	WT	SVM	98.4
Mai et al. 2011 [30]	Unique	1	Stance	Mech	Mean, Force Changing Rate, Force Ratio	ANN	98.5
Mai et al. 2018a [31]	Sliding	NA	100	Mech	Mean, Max, Min, SD, Diff	SVM	NP
Mai et al. 2018b [32]	Sliding	NA	100 pts	Mech	Mean, Max, Min, SD, Diff	SVM	99.4
Miller et al. 2013 [33]	Multiple	3	200/300/100	EMG	MAV, SSC, WL, ZC, SD	SVM	98.5
Moon et al. 2019 [34]	Sliding	NA	NP	Mech	Raw data	ANN	NP
Pew et al. 2017 [35]	NP	NP	NP	Mech	NP	KNN	93.8
Shell et al. 2018 [36]	Sliding	NA	150	Mech	Mean, SD, Max, Min	LDA	78
Simon et al. 2017 [37]	Multiple	2	300	Mech	WT	DBN	99.6
Spanias et al. 2014 [38]	Multiple	2	300	EMG Mech	MAV, SSC, WL, ZC, AR Mean, Max, Min, SD, IV, FV	LDA	~ 96
Spanias et al. 2015 [39]	Multiple	8	300	EMG Mech	MAV, SSC, WL, ZC, AR Mean, Max, Min, SD, IV, FV	DBN	~ 99
Spanias et al. 2016a [40]	Multiple	2	300	EMG Mech	MAV, SSC, WL, ZC, AR Mean, Max, Min, SD	DBN	NR
Spanias et al. 2016b [8]	Multiple	8	300	Mech	Mean, Max, Min, SD, IV, FV	DBN	96.7
Spanias et al. 2017 [41]	Multiple	8	300	Mech	Mean, Max, Min, SD, IV, FV	DBN	98.8
Spanias et al. 2018 [42]	Multiple	4	300	EMG Mech	MAV, SSC, WL, ZC, AR Mean, Max, Min, SD, IV, FV	DBN	95.97
Stolyarov et al. 2017 [43]	Unique	1	FF to FO	Mech	Mean, Max, Min, SD	LDA	94.1
Su et al. 2019 [44]	Unique	1	490	Mech	Raw Data	CNN	89.2
Tkach et al. 2013 [45]	Multiple	3	250	EMG Mech	MAV, SSC, WL, ZC Mean, SD	LDA	96
Wang et al. 2013 [46]	Multiple	4	200	Mech	Range, AR, CORR	LDA	99.01
Wang et al. 2018 [47]	Sliding	NA	50 pts	Mech	Raw Data	LSTM	95
Woodward et al. 2016 [48]	Multiple	2	300	Mech	Mean, Max, Min, SD, IV, FV	ANN	98.9
Xu et al. 2018 [49]	Sliding	NA	250	Mech	Mean, Max, Min, SD, Diff	QDA	93.2
Young et al. 2013a [50]	Multiple	8	300	EMG Mech	MAV, SSC, WL, ZC, AR Mean, Max, Min, SD	DBN	~ 98.2
Young et al. 2013b [51]	Multiple	8	300	Mech	Mean, Max, Min, SD	DBN	~ 98
Young et al. 2013c [52]	Multiple	2	300	EMG Mech	MAV, SSC, WL, ZC, AR Mean, Max, Min, SD	LDA	86.4
Young et al. 2014a [53]	Multiple	2	250	Mech	Mean, Max, Min, SD	LDA	~ 99
Young et al. 2014b [54]	Multiple	8	300	EMG Mech	MAV, SSC, WL, ZC, AR Mean, Max, Min, SD	DBN	~ 99
Young et al. 2016 [55]	Multiple	8	300	Mech	Mean, Max, Min, SD, IV, FV	DBN	~ 99
Zhang et al. 2011 [56]	Sliding	NA	150	EMG Mech	MAV, SSC, WL, ZC Mean, Max, Min	LDA	> 97
Zhang et al. 2013 [57]	Sliding	NA	150	EMG Mech	MAV, SSC, WL, ZC Mean, Max, Min	SVM	95
Zhang et al. 2019 [58]	Sliding	NA	600	Depth Camera	Raw data	CNN + HMM	96.4
Zhang et al. 2019 [59]	Sliding	NA	733	Depth Camera	Raw data	CNN	94.9
Zhang et al. 2012 [60]	Sliding	NA	160	EMG Mech	MAV, SSC, WL, ZC Mean, Max, Min	LDA	97.6
Zheng et al. 2013 [61]	Sliding	NA	250	Capacitive	Mean, Max, Min, SD, sum(abs(diff(X))), mean(diff(X)), sum(abs(X)), Std(abs(diff(X))), CORR	QDA	95
Zheng et al. 2014 [62]	Sliding	NA	250	Capacitive	Mean, Max, Min, SD, sum(abs(diff(X))), mean(diff(X)), sum(abs(X)), Std(abs(diff(X))), CORR	QDA	95.1
Zheng et al. 2016 [63]	Sliding	NA	250	Capacitive Mech	Mean, Max, Min, SD, sum(abs(diff(X))), sum(abs(X)) Mean, Max, Min, SD	SVM	95.8
Zheng et al. 2019 [64]	Sliding	NA	250	Mech	Mean, Max, SD	SVM	92.7
Zhou et al. 2019 [65]	Sliding	NA	150	Mech	Mean, Max, Min, SD, RMS	SVM	>90

**Analysis Windows: Type**: Three types of analysis windows are used: sliding windows, multiple windows, unique window. **Number:** When using multiple windows, the number of windows is reported. NA = Not Applicable (for unique and sliding windows). NP = Not Provided. **Length:** The window length is reported in ms. When the window length is variable, both the beginning and the end of the window(s) are reported: Gait Cycle = the data of the complete gait cycle are extracted, Stance = The data of the stance phase of the tested side are extracted, FF to FO = the data from Foot Flat to Foot Off (of the tested side) are extracted, FC contro to FC ipsi = the data from the Foot Contact of the contralateral side to the Foot Contact of the ipsilateral side are extracted. If the acquisition frequency was not reported, the window length is reported in terms of point numbers. NP = Not Provided. **Sensors:** EMG = ElectroMyoGraphs, FMG = ForceMyoGraph, Mech = Mechanical sensors (e.g., IMU, joint angle, joint rotational speed, data from load cell, etc.), for more details refer to Table 3. **Features:** NP = Not Provided, WT = Wavelet Transform, DTW = Dynamic Time Wrapping, Max = Maximum value, Min = Minimum Value, IV = Initial Value, FV = Final Value, RMS = Root Mean Square, SD = Standard Deviation, WL = Waveform Length, ZC = Zero Crossings, SSC = Slope Sign Change, MAD = Mean Absolute Deviation, Diff = Differential Values, AR = autocorrelation coefficients of an autoregressive model (the number of coefficients and the order of the model are not reported), CORR = Correlation between signals, ICC = Intraclass Correlation Coefficients. **Algorithm**: (by order of appearance) SVM = Support Vector Machine, HMM = Hidden Markov Model, LDA = Linear Discriminant Analysis, LR = Logistic Regression, EBA = Entropy Based Algorithm, CNN = Convolutional Neural Network, ANN = Artificial Neural Network, DT = Decision Tree, LIFT = Learning From Testing Data, KNN = K-Nearest Neighbor, DBN = Dynamic Bayesian Network, LSTM = Long-Short Term Memory network, QDA = Quadratic Discriminant Analysis. **Accuracy**: NP = Not Provided, NR = Not Reported (in Huang et al. [23] and in Spanias et al. [40], the influence of simulated noise on the EMG was tested, the reported accuracies were lower and were not comparable to other studies), A ‘~’ sign means that results were obtained from reading graphs un the paper.

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
