# Peer review of "Machine Learning Approaches for Activity Recognition and/or Activity Prediction in Locomotion Assistive Devices—A Systematic Review"

_sensors, 2020, doi:10.3390/s20216345_

Round 1

Reviewer 1 Report

This paper presents a comprehensive systematic review of the literature (2000-2020) reporting the use of machine learning (ML) to classify locomotion modes in lower-limb wearable assistive devices. An ensemble of 76 articles meeting all the imposed constraints were analyzed and key features such as: QS, type of device, locomotion activities, sensors, ML algorithms, their accuracy, etc. were extracted for comparison. The discussion well summarizes the findings and the paper gives some general recommendations for future similar studies.

In sum, this work proposes an interesting review and assessment of papers addressing a particular topic on ML applied to rehabilitation. In my opinion, the study was well conducted, formally describes all concepts involved, and is technically sound. Moreover, the paper is well written, it is easy to read, and to understand.

Yet, I recommend a major revision with the following remarks:

1) I was surprised the authors did not considered Scopus, the largest database of scientific literature. Roughly, it offers 5 times more bibliographical entries than the databases reported in the study (PubMed and WoS).

2) After reviewing for several years for MDPI, I can say that the abstract’s format is wrong: Background-Method-Conclusion. The abstract has to be in a narrative format.

3) Line 78: Please provide the reference to your work in PROSPERO. Ref. [6] does not point to it.

4) Line 176: 1,267 exclusions from 1,343 candidates are indeed striking. The 94% of the papers in the literature did not meet your selection criteria!

5) Line 179: Figure 1, Where is it? There is no figure 1 neither in the paper nor in the supplementary material (4 files).

6) Besides from (alleged) figure 1, the paper does not contain any other figure. I consider that the explanation of the four “Critical Timing” (section 3.3.3) approaches merits one, from Huang to Xu. By the way, the second (Line 244) and third (Line 248) ones do not point to any references.

7) Minor: all over the paper (starting from the title), you use “assisting devices”. I believe it is: “assistive devices”.

Author Response

All of the authors thank the reviewer for his comments. They will serve to improve our paper. Please find in the document attached our replies to the questions raised. We also attached a revided version of the manuscript after our responses.

Reviewer 2 Report

Summary of Contributions

The paper presents the systematic review of numerous studies in which machine learning has been used to adapt the behavior of orthotic/prosthetic devices to user locomotion mode. In detail, it included/excluded in any condition in the article published for 20 years, and evaluated the article with the QualSyst tool. And it investigated the sensors used in each article and analyzed the algorithms they used.

Strengths
1) Information used for many articles, such as sensors and locomotion activities, was investigated.
2) Based on the investigations, the analysis of the relation between sensors and locomotion mode is included.

Weaknesses
1) In section 2.4.2, there is a lack of explanation about the standard of information provision used when the article was rated with a score between 0 and 2.
2) In table 1, because the number of recruited volunteers is not all compared to the same value, it is difficult to compare each experiment and the number of that is also small.

Author Response

(The authors gave the same response as above.)

Round 2

Reviewer 1 Report

The manuscript has undoubtedly improved from its previous version. I appreciate that my remarks were all taken into account. In particular, new figure 2 clarifies the critical timings discussed in section 3.3.3. Together with the precisions added to this version, the modifications made strengthen the paper.

I have no further comments or suggestions; therefore, I now recommend this paper’s acceptance.

Reviewer 2 Report

- In this revision, the author well complemented my concerns.

- There are no major grammatical errors, but I recommend that the paper is proofread before it is published.